# Biomechanical Aspects of Various Attachments for Implant Overdentures: A Review

**DOI:** 10.3390/polym13193248

**Published:** 2021-09-24

**Authors:** Bharat Mirchandani, Ting Zhou, Artak Heboyan, Sirasa Yodmongkol, Borvornwut Buranawat

**Affiliations:** 1Department of Periodontics & Implant Dentistry, Faculty of Dentistry, Thammasat University, Patum Thani 12120, Thailand; drbharat@staff.tu.ac.th; 2School of Stomatology, Kunming Medical University, Kunming 650051, China; zhouting737373@gmail.com; 3Department of Prosthodontics, Faculty of Stomatology, Yerevan State Medical University after Mkhitar Heratsi, Str. Koryun 2, Yerevan 0025, Armenia; heboyan.artak@gmail.com; 4Assistive Technology and Medical Devices Research Center (A-MED), National Science and Technology Development Agency (NSTDA), Pathum Thani 12120, Thailand; sirasa.yod@nstda.or.th

**Keywords:** dental implant, overdenture, attachments, polymethyl methacrylate, stress, retention

## Abstract

There have been considerable recent technological developments for implant overdenture attachments. This study presents an overview of the biomechanical and biomolecular aspects of various attachments for implant overdenture. Available articles on attachments for implant overdenture were reviewed from January 1980 to August 2021 in the ScienceDirect, MEDLINE/PubMed, and Web of Science resources, and relevant studies were included in this study. We focused on the following topics: attachment systems, retention of various attachments, stress distribution with different attachments, the design and fabrication of attachments, digital techniques in overdenture attachments, and the effects of attachments in peri-implant health. We found that plastic resin is commonly used for ball and bar attachments, whereas nylon resin is commonly used in locator attachments. The locator system offers a valuable attachment option for implant-retained overdenture. Attachment retention reduces while lateral force increases with implant inclination in overdenture. The higher the retention of an overdenture attachment, the higher the transferred stresses. Additionally, clip loading produces more stress in implants and precision elements than bar-retained dentures. As such, we conclude that the ball and locator systems the best overdenture systems due to their superior tissue response, survival rate, and patient satisfaction.

## 1. Introduction

At present, implants are widely used to replace missing teeth or retention/support dentures [1]. The use of implant-retained overdentures in the maxilla and mandible is a successful option to the fixed implant prostheses. The types of attachments available in the market include non-splinted attachments (ball, magnet, locator, and double crown attachment) and splinted attachments (bar and clip attachment) [2,3,4]. Figure 1 shows the ball attachment to retain overdenture [3].

The locator attachment system is a suitable choice for implant-retained or implant-supported overdenture [5]. There has been considerable development in attachments for implant overdenture; however, there is a need for updated information regarding implant overdenture attachments is lacking. As such, this review presents an overview of various attachments’ biomechanical and biomolecular aspects, with focus on attachment systems, the retention of various attachments, stress distribution with different attachments, the design and fabrication of attachments, digital techniques in overdenture attachments, and the effects of attachments in peri-implant health. After assessing the available articles on overdenture attachments from January 1980 to August 2021 using the PubMed/MEDLINE, ScienceDirect, and Web of Science resources, relevant studies on overdenture attachments were included in this study.

## 2. Polymeric Materials Used for Overdenture

Various polymeric materials are used in overdenture attachments (as shown in Table 1), including polyvinylsiloxane (PVS) [6], polymethyl-methacrylate (PMMA) [7], plastic resin, Hostaform (polyoxymethylene copolymer) (POM) [8], nylon resin, and DuPont Zytel [8]. Plastic resin is commonly used for ball and bar attachments, whereas nylon resin is commonly used in locator attachments.

## 3. Effects of Attachment Types on Implant Overdenture

Attachments vary in shape and include ball, bar, locator, magnet, and locator types. Stud attachments are either rigid (ball) or resilient (magnet, locator, and double crown attachment). Table 2 presents an overview of the various attachments in overdenture.

The ball attachment (O-ring attachment) is the most commonly used overdentures and contains a ball shape for retention. Its advantages include a simple manufacturing process, the provision of a wide range of movement, cost-effectiveness, ease of use and maintenance, the provision of good retention, hygiene maintenance, and good patient satisfaction [10,11]. However, the ball attachment abutment requires implants to be parallelly placed, and the loss of parallelism may cause difficulty while inserting and removing the prosthesis or during the fracturing of the abutment [10]. Additionally, the O-ring needs to be regularly changed because it is subject to wear [10].

A bar attachment offers retention, the splinting of implants, and the distribution of load, resulting in reduced implant stress, which is critical for the immediate loading protocol [12]. The restoration of moderately to severely atrophic maxilla remains a challenge. In such cases, CAD/CAM titanium bar-supported overdenture can be an important treatment choice for an edentulous patient’s rehabilitation (Figure 2) [22]. Figure 2 shows a maxillary overdenture supported by four or six implants for the minimally invasive rehabilitation of atrophic maxillae. The disadvantages of bar attachments include technique-sensitivity, high costs, and difficult hygiene maintenance under the bars that leads to mucosal swelling or gingival hyperplasia [12,13]. Furthermore, bars are not indicated in a V-shaped ridge because this causes the infringement of tongue space [23].

Locators are currently popular attachments because of their low level of thickness (2.5 mm height) [16] and ability to self-align, which can correct up to 40° of implant angulations [16]. They can be used in narrow inter-arch space and prevent the fracture of the denture base [17]. Locators offer excellent retention and stability, and they allow for easy hygiene maintenance. The telescopic attachment, which offers a self-seating mechanism, is appropriate for patients with reduced manual dexterity, such as those with Parkinson’s disease. However, the periodic replacement of the male nylon part is required [14]. Some prosthetic complications have been noted in locator attachments. One study reported 34 prosthetic complications and a locator housing requiring 16 replacements [8,15]. To avoid complications, locator attachments require periodic repair and higher maintenance [24]. Recently, researchers invented a double-crown attachment option for locator attachments have that connects dentures to prepared teeth [25]. However, the disadvantages of locators include the need for sufficient inter-arch space and the metal display of attachments [26]. The locator attachment system is a suitable choice for implant-retained or implant-supported overdenture [5].

Magnetic attachments have a long history (>60 years) of use in denture retention [27]. They reduce the transfer of horizontal stress to the implants and the bone during the insertion and removal of the denture [18]. They are low-profile attachments [19], and the corrosion and loss of magnetism are significant complications associated with their usage [20,21].

Overdenture attachments present very high survival rates. One study reported survival rates ranging from 96% to 97% for bar attachments, 96% to 100% for ball attachments, 90% to 92% for magnets, and 97% for locators with a mean follow-up period of 3 years [24]. Other studies have reported a 94% five-year survival rate for the bar attachment [28]; 89% and 93% survival rates for the bar and locator, respectively; [29], and 98% and 97% survival rates for the bar and the locator, respectively [30].

## 4. Retentive Force and Stress Distribution of Various Attachments

Retentive force is vital for the success of overdenture. It is essential to consider the biomaterial aspects of attachments when choosing the appropriate attachment system for overdenture.

The retentive stress and strain absorbed during the removal of stud attachments in implant-supported overdenture are important criteria for selecting attachments [31,32]. Petropoulos and Mante [31] compared the retention of six types of attachments in vitro implant-retained overdenture. They reported that the Zest Anchor Advanced Generation attachment had the greatest retentive vertical force (37.2 N) and oblique force (25.9 N). They reported the lowest vertical force (10.8 N) and oblique retentive force (10.6 N) for the Zest Anchor and Nobel Biocare Standard attachments. The Nobel Biocare Standard Ball attachment was found to have the highest strain in the vertical and oblique directions, while the Zest Anchor and Sterngold-implanted ERA White attachment had the lowest vertical and oblique strain energies. Similarly, the authors of another study reported that the mean retention for ball attachments was better than that of locator attachments (difference of 5.0 N), and patients preferred ball attachments over locator attachments [33]. Hence, in the vertical and oblique directions, the Zest Anchor and Nobel Biocare Standard attachments present the lowest strain energies and ball attachments present the highest strain energies.

Finite element analysis (FEA) is commonly used in stress analysis [34,35,36]. An in vitro study by Idzior-Haufa et al. [37] compared the biomechanical properties of two bar-retained implants and two bar-retained implants with ball attachments for overdenture (Figure 3). They performed FEA analysis for five modes of loads: 20, 50, and 100 N of vertical force and 20 and 50 N of oblique force exerted on individual teeth (central incisor, canine, and first molar). The maximum bar and implant stress values were found to be 27.528 and 23.16 MPa, respectively. The values of the maximum stress on the bar, clips, and implants were 578.6, 136.99, and 51.418 MPa, respectively. The authors stated that the clips’ loading produced higher stress in the implants and precision elements compared to the bar-retained denture.

Similarly, Yang et al. [38] studied the retention and lateral force of an implant with respect to its inclination with four types of attachments for overdentures under a constant dislodging force and implant angles of 0°, 15°, 30°, and 45°. The authors reported that the order of maximum retention was as follows: locator blue, locator black, ball attachment, flat-type, and self-adjusting magnetic attachments. They found that (except for magnetic attachments) with implant inclination, retentive force decreases and lateral force increases. Elkerdawy et al. [39] evaluated the stress pattern and retention of a ball attachment and two telescopic attachments of various convergence angles in implant-supported overdenture. The experiment was carried out in three conditions: with the ball attachment and with the telescopic retainers at angles of 4° and 6°. The authors found that the ball attachment presented the greatest retention, followed by the telescope angles of 4° and then 6°. Hence, the higher the retention of the overdenture attachment, the higher the transmitted stresses.

Arsos et al. [40] compared the retention and durability of three types of attachments: Dalbo-Plus®, Clix®, and Locator®. Samples were placed in their acrylic resin forms and subjected to mechanical testing (5400 cycles of insertion and removal) over the respective ball or locator abutments. The abutments were placed at 0°, 10°, and 20°. The authors found that there were significant differences in the average values of the insertion or removal force due to angulation and type of attachment. The greater angulation of the abutments influenced the retention of the attachments. A test intended to simulate five years of fatigue showed that denture insertion and removal did not cause wear in the metal abutments.

Finally, Porter et al. [41] compared the force distributions of implant overdenture attachments when vertical forces were applied; the Nobel Biocare standard ball (NSB), Nobel Biocare 2.25 mm diameter ball (NB2), Nobel Biocare bar and clip (NBC), Zest Anchor Advanced Generation (ZAAG), Sterngold ERA Orange (SEO), Sterngold ERA White (SEW), Compliant Keeper System with titanium shims (CK-Ti), Compliant Keeper System with clear silicone 2SR90 sleeve rings (CK-90), and Compliant Keeper System with black nitrile 2SR90 sleeve rings (CK-70) were used, and static loads were applied (100 N) (1) bilaterally over the distal midline (DM), (2) unilaterally over the right implant (RI), (3) unilaterally over the left implant (LI), and (4) between implants in the mid-anterior region (MA). The authors found that both the loading location and attachment type were statistically significant factors. The force and stress on implant were greater when the load was applied over the implant or at the MA. While not significant at every loading location, the largest implant forces occurred with the ZAAG attachments, and the smallest forces were found with the NSB, SEO SEW, CK-70, and CK-90 attachments. More movement was observed for the NBC and ZAAG attachments, and lossless movement was observed for the SEO, NSB, SEW, CK-90, and CK-70 attachments.

## 5. Effects of Attachments in Peri-Implant Health

Various prosthetic factors and attachments have impacts on peri-implant health [24,42]. Aldhohrah et al. [2] reviewed the effect of the current attachment systems used in two-implant-retained mandibular overdenture (including bar, ball, locator, resilient telescopic, and magnet attachments) on peri-implant tissue health; they reported that all attachment systems had the same effect on marginal bone loss and probing depth. Moreover, Chaware and Thakkar [24] reported that maintaining peri-implant tissue health is more difficult for the bar attachment, which showed moderate gingival inflammation and bone resorption, than other attachments.

A crossover clinical trial found that the ball attachment leads to excellent peri-implant health, with a significantly lower tissue response than bar attachments [43]. In addition, the magnetic attachments initially showed a significantly high plaque index and the bar attachments showed a rise in gingival inflammation after 1.5 years. Thus, ball attachments present the best peri-implant tissue health compared to other attachments [44,45]. Similarly, the marginal bone loss after 1 and 2 years for locator attachments was found to be 0.58 ± 0.71 and up to 6 mm, respectively, and the marginal bone loss for bar attachments after 1 and 2 years was found to be 0.31 ± 0.47 and up to 10 mm, respectively [29,30].

For the immediate-loading protocol, non-splinted attachments (bar, ball, and magnet) are the best overdenture attachments due to their resiliency and bone loading within physiological limits. Furthermore, with delayed loading, there may be increased bone loss due to trauma associated with second-stage surgery. However, a disadvantage of non-splinted implants that retain mandibular overdentures is that they are associated with higher bone loss than splinted implants (bar attachment) [46] due to the splinted bar resulting in wide load distribution and reducing implant micromotion and crestal bone loss.

Assad et al. [45] compared a mandibular magnet-retained overdenture (mainly mucosa-supported) and a bar-retained overdenture (combined mucosa-implant-supported). The authors concluded that the magnet-retained overdenture has less bone resorption distal to an implant than a bar-retained overdenture. Magnet-retained overdentures were found to be associated with a high plaque index score. After a follow-up of 1.5 years, the bar-retained overdenture showed a significant increase in gingival inflammation. In addition, in a systematic review and meta-analysis by Keshk et al. [47] on telescopic crown and ball attachments for mandibular implant overdentures found no significant differences regarding marginal bone loss, plaque index, bleeding index, gingival index, and prosthodontic maintenance between the two groups.

## 6. Effects of Attachments on Patient’s Satisfaction and Quality of Life

Implant overdentures’ success depends on a predictable attachment system that connects the implants to the prosthesis [24,33,48]. Rosa et al. [48] compared patient satisfaction with various attachment systems for overdenture. They reported no differences in patient satisfaction between the ball and the other attachment systems. The ball attachment was only found to be superior to the magnet attachment. Another study reported the following increasing order for patient compliance and satisfaction: ball, locator, bar, and magnet attachments [24].

De Albuquerque et al. [33] studied and compared the retention and patient outcomes of the ball and cylindrical attachment systems for implant overdentures. They found that the mean retention time was greater for the ball attachment than the cylindrical attachment. For both attachments, the retention considerably decreased over time, though this began earlier for the cylindrical systems than the ball systems. Patient satisfaction for ball and conus-retained implant overdentures was 64% and 100%, respectively [49].

Brandt et al. [50] compared locator and ball attachments systems regarding prosthetic maintenance and patients’ oral-health-related quality of life (OHRQoL) by utilizing Oral Health Impact Profile-G14 (OHIP-G14). Kaplan–Meier survival analysis (Figure 4) showed significant favor for locators regarding patrix-related events (*p* = 0.030) (Figure 4A). Similarly, locators were found to be better in relation to matrix-related events (*p* = 0.028) (Figure 4B) and denture fracture (*p* < 0.001) (Figure 4C). The locator group had a considerably lower OHIP-G14 score than the ball attachment group (*p* = 0.002). The ball attachment group needed more maintenance than the locator group. Additionally, patients with locators had better OHRQoL scores than patients with ball attachments.

## 7. Recent Advances and Future Perspectives

Research and clinical applications in implant dentistry had led to the development of various bio and digital prosthetic dentistry materials [51,52,53,54,55]. A key developmental component has comprised advances in artificial intelligence (AI), which has been implemented in several dental and dental technology workflows, especially that of CAD/CAM [53].

Newer materials can be integrated with overdenture attachment systems. Recently, polyether ether ketone (PEEK) and polyether ketone ketone (PEKK) have been widely used in implant and restorative dentistry [56,57]. Li et al. [6] evaluated the retention of PEEK post-core restoration with polyvinylsiloxane (PVS) attachment systems (Figure 5), and their cyclic dislodgement test showed an inverse linear relationship between cyclic times and retention force. The PVS’s retention was enhanced with an increase in Shore hardness, thus showing a favorable retention force. Therefore, post-core PEEK with PVS attachments may comprise an excellent alternative attachment system in dentistry.

Digital dentistry is rapidly developing and is not limited to provisional restorations or implants; rather, it permeates all aspects of this profession. However, existing digital dentistry research and applications have limitations. It has been shown that dentists are using only a fraction of available data for planning and treatment, so they are not fully utilizing the growing body of information [53]. Additionally, numerous studies have not entirely engaged with the rigorous planning, conducting, and reporting standards established by evidence-based research practice. There is a need to integrate this technology, clinical dentistry, and interdisciplinary research to overcome these issues.

## 8. Conclusions

We found that the locator system offers a valuable attachment option for implant-retained overdenture. Attachment retention reduces while lateral force increases with implant inclination in overdenture. The higher the retention of the overdenture attachment, the higher the transferred stresses. In the vertical and oblique directions, the Zest Anchor and Nobel Biocare Standard attachments present the lowest strain energies and the ball attachments show the highest strain energies. Clip loading produces more stress in implants and precision elements than bar-retained dentures. As such, we conclude that the ball and locator systems are the best overdenture systems due to their superior tissue response, survival rate, and patient satisfaction. Carefully maintaining an attachment system with prostheses and mucosa is important for peri-implant health.

## Figures and Tables

**Figure 1 polymers-13-03248-f001:**
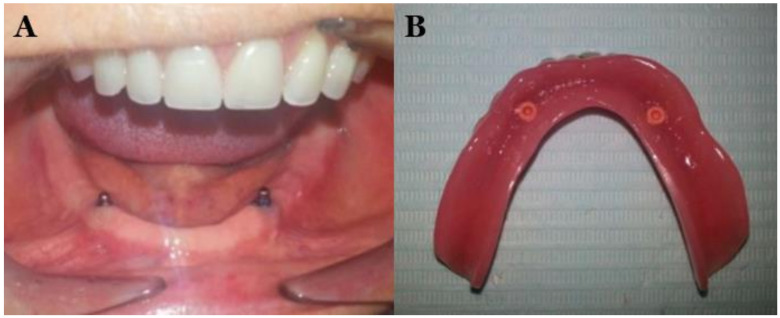
Ball attachment to retain overdenture (**A**) and overdenture (tissue surface) (**B**) [3].

**Figure 2 polymers-13-03248-f002:**
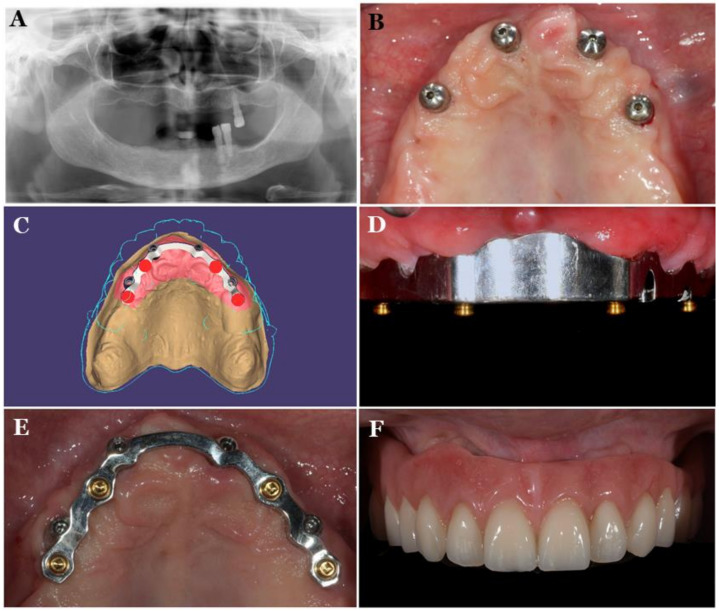
CAD/CAM titanium-bar-supported maxillary overdenture Pretreatment radiograph (**A**), 4 implants placed in maxillary arch (**B**), CAD design of the titanium bar (**C**), titanium bar inserted in the mouth (**D**), occlusal view of titanium bar (**E**), definitive prosthesis (**F**) [22].

**Figure 3 polymers-13-03248-f003:**
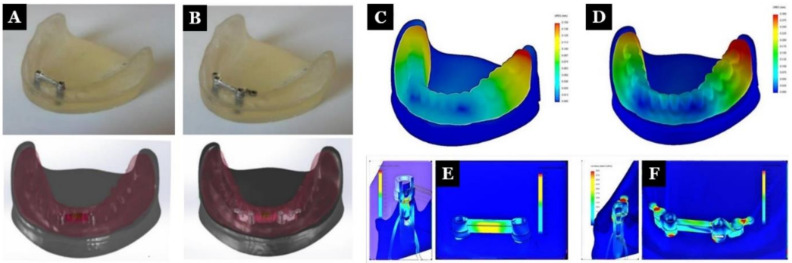
Real and computer models of the first studied system I (**A**) and the second studied system II (**B**). Distribution of displacements in an overdenture when an incisor was loaded with a vertical force of 50 N in system I (**C**) and system II (**D**). Distribution of stress within an implant and precision element when an incisor was loaded with a vertical force of 50 N in system I (**E**) and system II (**F**) [37].

**Figure 4 polymers-13-03248-f004:**
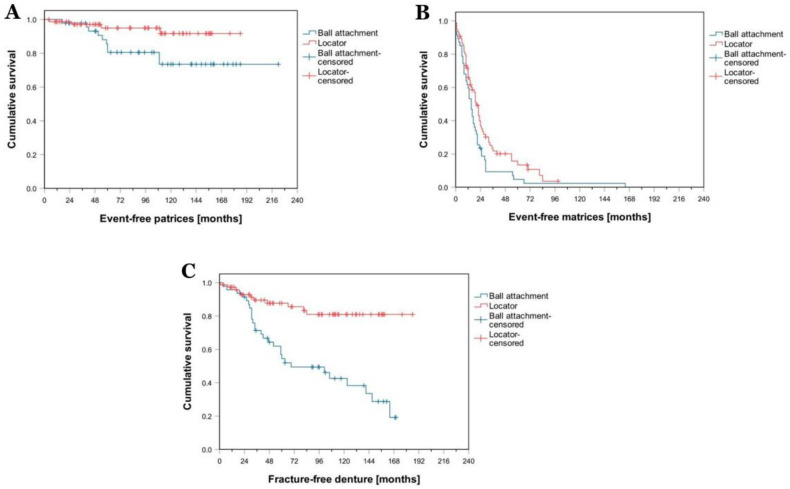
Results of Kaplan–Meier survival curves regarding patrix-related events (**A**), matrix-related events (**B**), and denture fracture (**C**) for locator and ball attachments [50].

**Figure 5 polymers-13-03248-f005:**
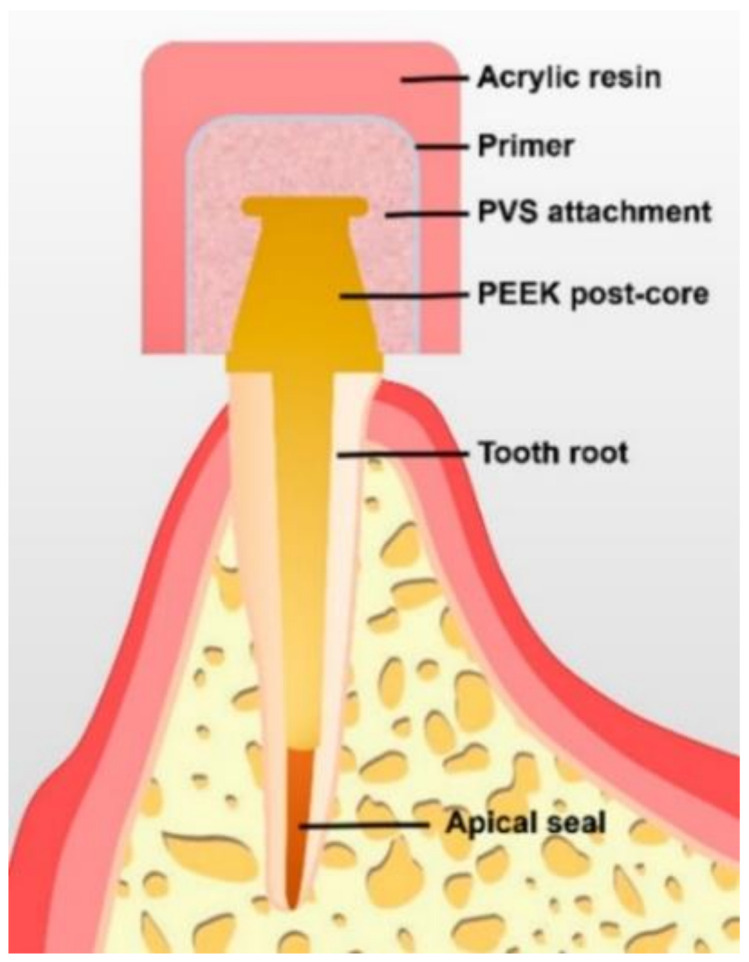
Diagrammatic representation of polyetheretherketone post-core restoration with a polyvinylsiloxane attachment system [6].

**Table 1 polymers-13-03248-t001:** Various polymeric materials used for attachment systems in overdenture.

Polymeric Materials	Structure	Advantages	Disadvantages	Reference
Polyvinylsiloxane (PVS)	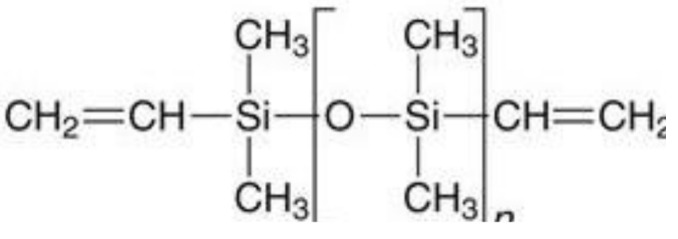	Presents stable retention and is economical.	PVS attachments result in distortion from repeated wear and artificial saliva.	[6]
Polymethyl-methacrylate (PMMA)	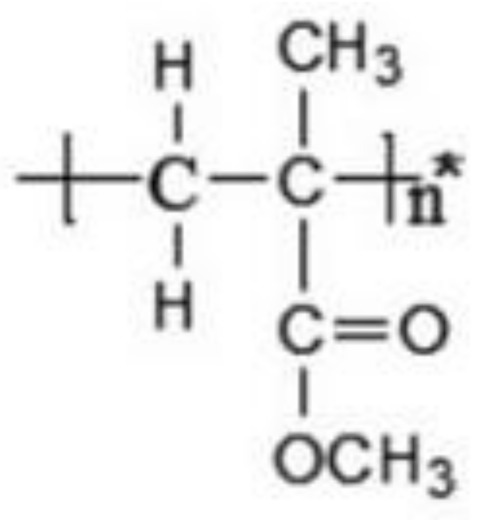	Durable, strong, aesthetic, presents good marginal adaptation, and is economical.	Exothermic polymerization, polymerization shrinkage, poor wear resistance, and pulp irritation associated with excess monomers.	[7,9]
Plastic resin and Hostaform (polyoxymethylene copolymer) (POM)	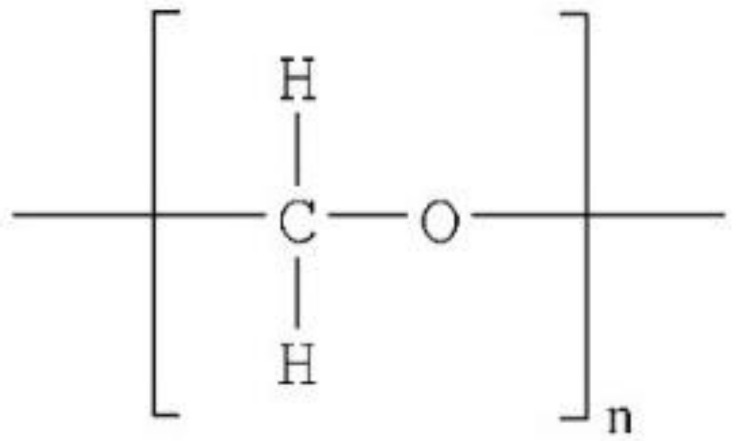	Elastic, aesthetic, presents good marginal adaptation, and is economical.	Minimal wear is seen in plastic matrices with minimal maintenance.	[8]
Nylon resin and DuPont Zytel	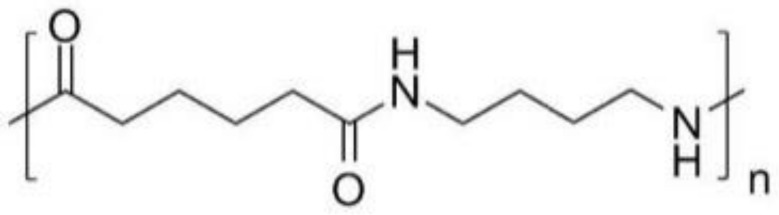	Elastic, aesthetic, presents good marginal adaptation, and is economical.	Locator nylon matrices show extensive deformation and require considerable maintenance.	[8]

**Table 2 polymers-13-03248-t002:** Advantages and disadvantages of the various attachments in overdenture.

Attachment Type	Advantages	Disadvantages	Reference
**Ball attachment** **(O-ring attachment)**	Simple manufacturing process, the provision of a wide range of movement, cost-effectiveness, ease of use, good retention, easy hygiene maintenance, and high patient satisfaction.	The abutment requires implants to be parallelly placed, and the loss of parallelism may cause difficulty while inserting and removing the prosthesis or during the fracturing of the abutment. The O-ring needs to be regularly changed because it is subject to wear.	[10,11]
**Bar attachment**	Provides retention, implant splinting, and wide-ranging load distribution that results in a movement reduction of the implants.	Technique-sensitive, expensive, and present difficult hygiene maintenance under the bars, leading to mucosal swelling or gingival hyperplasia. Bars are not indicated in a V-shaped ridge because this leads to the infringement of tongue space.	[12,13]
**Locators**	Locators are popular attachments for implant-retained or implant-supported overdenture because of their low level of thickness (2.5 mm height) and ability to self-align, which can correct up to 40° of implant angulations. They can be used in narrow inter-arch space. Locators offer excellent retention and stability, and they allow for easy hygiene maintenance. The telescopic attachment, which offers a self-seating mechanism, is suitable for patients with reduced manual dexterity, such as those with Parkinson’s disease.	Periodic replacement of the male nylon component is required. Some prosthetic complications such as locator attachments, periodic repair, and higher maintenance double-crown locator attachments require sufficient inter-arch space and the metal display of attachments.	[8,14,15,16,17]
**Magnetic attachments**	Magnetic attachments reduce the transfer of horizontal stress to the implants and bone during the insertion and removal of the denture.	These are low-profile attachments; however, corrosion and loss of magnetism are significant complications associated with their usage.	[18,19,20,21]

## Data Availability

The data presented in this study are available on request from the corresponding authors.

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
