# Peer review of "Biomechanical Aspects of Various Attachments for Implant Overdentures: A Review"

_polymers, 2021, doi:10.3390/polym13193248_

Round 1
Reviewer 1 Report
It is a well written review, and could be accepted after minor revision. Please consider the following comments when you revise it.
- Table 1. There are two advantages in the Table, I think one of it should be disadvantages.
- I think Table 3 is unnecessary, put all the content in the paragraph is enough.
Author Response
Response to reviewer 1 comments
Thank you for your positive comments. Corrections in the Manuscript for Reviewer 1 are highlighted in Yellow color.
It is a well written review and could be accepted after minor revision. Please consider the following comments when you revise it.
Point 1: Table 1. There are two advantages in the Table, I think one of it should be disadvantages.
Response 1: There was mistake. It is corrected.
Point 2: I think Table 3 is unnecessary, put all the content in the paragraph is enough.
Response 2: Table 3 is deleted and added the contents in the paragraph.
Reviewer 2 Report
Comments on the paper:
The manuscript titled: "Biomechanical Aspects of Various Attachments for Implant Overdentures: A Review" is well written with valuable scientific information presented.
However, there are a few minor corrections that should be made.
1) Page 1, 3rd sentence, ")" is missing
2) Page 2, Table 1, please correct the structural formula for polymethyl-methacrylate (PMMA) from monomeric form to polymeric form (as the other polymeric forms in Table 1 are presented); also please correct the name from Polymetha-methacrylate to Polymethyl-methacrylate.
3)Please, correct in the same Table 1, in Polymeric materials column, "(polyoxymethyle ne copolymer) " to "(polyoxymethyle-ne copolymer) " i.e. insert "-" after "polyoxymethyle"
4) Page 4, in the paragraph under the Figure 2 please correct in the 1st sentence "40 ͦ " to"40°"
5) Page 5, Table 3, in row Yang et al. in Results column, at the end of the 2nd sentence "...except for in magnetic attachments" please omit "in"
6) Page 6, Figure 3, I suggest a small change in the title of the Figure 3, to insert "I" after "...of the first studied system..." and "II" after "...of the second studied system..." to get "...of the first studied system I (A) and of the second studied system II (B)."
7) Page 8, in Recent Advances and Future Perspective, 2nd paragraph, 2nd sentence, please correct the "polyetheretherketone" to "polyether ether ketone". Please, do the same in Figure 5 title.
Author Response
Response to reviewer 2 comments
Correction in the Manuscript are highlighted in Green color.
The manuscript titled: "Biomechanical Aspects of Various Attachments for Implant Overdentures: A Review" is well written with valuable scientific information presented.
However, there are a few minor corrections that should be made.
Point 1: Page 1, 3rd sentence, ")" is missing
Response 1: Corrected.
Point 2: Page 2, Table 1, please correct the structural formula for polymethyl-methacrylate (PMMA) from monomeric form to polymeric form (as the other polymeric forms in Table 1 are presented); also please correct the name from Polymetha-methacrylate to Polymethyl-methacrylate.
Response 2: Formula and name are corrected.
Point 3: Please, correct in the same Table 1, in Polymeric materials column, "(polyoxymethyle ne copolymer) " to "(polyoxymethyle-ne copolymer) " i.e. insert "-" after "polyoxymethyle"
Response 3: Name is corrected.
Point 4: Page 4, in the paragraph under the Figure 2 please correct in the 1st sentence "40 ͦ " to"40°"
Response 4: Corrected.
Point 5: Page 5, Table 3, in row Yang et al. in Results column, at the end of the 2nd sentence "...except for in magnetic attachments" please omit "in"
Response 5: Table 3 is removed as suggested by Reviewer 1. Removed in the text.
Point 6: Page 6, Figure 3, I suggest a small change in the title of the Figure 3, to insert "I" after "...of the first studied system..." and "II" after "...of the second studied system..." to get "...of the first studied system I (A) and of the second studied system II (B)."
Response 6: Corrected.
Point 7: Page 8, in Recent Advances and Future Perspective, 2nd paragraph, 2nd sentence, please correct the "polyetheretherketone" to "polyether ether ketone". Please, do the same in Figure 5 title.
Response 7: Corrected.